# Fabrication and Process Optimization of Chinese Fir-Derived SiC Ceramic with High-Performance Friction Properties

**DOI:** 10.3390/ma16124487

**Published:** 2023-06-20

**Authors:** Fuling Liu, Shanshan Chang, Yuanjuan Bai, Xianjun Li, Xiaojian Zhou, Jinbo Hu

**Affiliations:** 1College of Materials Science and Engineering, Central South University of Forestry and Technology, Changsha 410004, China; 20201100215@csuft.edu.cn (F.L.); changelxy@hotmail.com (S.C.); baiyuanjuan@csuft.edu.cn (Y.B.); lxjmu@163.com (X.L.); 2Yunnan Provincial Key Laboratory of Wood Adhesives and Glued Products, Southwest Forestry University, Kunming 650224, China; xiaojianzhou@hotmail.com

**Keywords:** fir-derived carbon, SiC ceramic, response surface optimization, high-performance friction

## Abstract

In this study, a novel friction material with biomass-ceramic (SiC) dual matrixes was fabricated using Chinese fir pyrocarbon via the liquid-phase silicon infiltration and in situ growth method. SiC can be grown in situ on the surface of a carbonized wood cell wall by mixing and calcination of wood and Si powder. The samples were characterized using XRD, SEM, and SEM–EDS analysis. Meanwhile, their friction coefficients and wear rates were tested to study their frictional properties. To explore the influence of crucial factors on friction performance, response surface analysis was also conducted to optimize the preparation process. The results showed that longitudinally crossed and disordered SiC nanowhiskers were grown on the carbonized wood cell wall, which could enhance the strength of SiC. The designed biomass-ceramic material had satisfying friction coefficients and low wear rates. The response surface analysis results indicate that the optimal process could be determined (carbon to silicon ratio of 3:7, reaction temperature of 1600 °C, and 5% adhesive dosage). Biomass-ceramic materials utilizing Chinese fir pyrocarbon could display great promise to potentially replace the current iron–copper-based alloy materials used in brake systems.

## 1. Introduction

Biomass materials have been demonstrated to be a valuable source of carbon due to their advantageous properties, such as damage tolerance, toughness, biocompatibility, and short growth cycle [1,2,3]. Recent studies have explored the potential of utilizing agricultural waste biomass as a single or common source of carbon and silicon for the preparation of ceramic materials with specific structural and functional properties for green manufacturing [1,4,5]. These studies suggest that the use of biomass materials could provide a low-carbon, cost-effective, and sustainable route for the production of ceramic materials [6,7]. Biomass-ceramic silicon carbide (SiC) dual matrix friction materials are a new generation of high-performance friction materials that have been developed to replace the current iron–copper-based alloy materials used in brake systems [8,9]. These materials have a complex manufacturing process, high density, high cost, brake noise, and a tendency to damage the couple disc, leading to increased wear and waste of resources. In addition, under emergency braking conditions, the maximum local surface temperature of the brake friction vice can reach 1100 °C, which is the limit of the use of conventional metal-based brake materials [10,11,12]. Furthermore, iron- and copper-based materials are prone to electrochemical corrosion in acidic rain and salt spray environments. In order to reduce dependence on non-renewable resources and energy, alleviate environmental degradation and resource shortages, and meet the needs of brake systems in high-speed, high-energy load conditions, the development and use of plant-based natural renewable resources is essential [13,14,15]. The new generation of high-performance and high-reliability friction materials not only has the basic requirements of current friction materials but also has high specific strength, long life, and environmental adaptability [9,16,17].

Silicon carbide (SiC) is a high-tech ceramic material bonded by covalent bonds that exhibits exceptional room-temperature mechanical properties, such as high flexural strength, excellent oxidation resistance, good corrosion resistance, and high wear resistance [18,19]. With an average bond energy between Si and C of 300 kJ/mol and a covalent to ionic bond ratio of 4:1, SiC ceramics are used in a variety of applications, including bearings, mechanical seals, nozzles, cutting tools, piston rings/cylinder liners in internal combustion engines, heat exchangers, and turbine components [20,21]. Direct carbonization of monolithic Si [22] and thermal reduction of SiO_2_ [23] by carbon are considered more desirable methods of producing SiC materials due to their low cost, ease of process, and high productivity [24]. However, chemical vapor deposition of SiC via precursors of SiH_4_ and small hydrocarbon molecules (e.g., C_2_H_4_, C_3_H_8_) diluted in a hydrogen carrier gas, processed at 1500~1600 °C, is limited due to the need for higher temperatures (up to 2700 °C) due to the low diffusion coefficient of the strong covalent Si–C bonding, the small contact area between reactants, and the low efficiency of heat transfer [25,26].

Biomass-based silicon carbide ceramics are a type of silicon carbide ceramic material [27]. Based on the structure of biomass, such as wood, bamboo, orange citrus, etc., using its own carbon and silicon elements and external silicon sources or other substances, high-temperature silicon infiltration in the wood template, the introduction of Si, SiO_2_, or organic precursors containing Si, in an inert atmosphere, through organic–inorganic transformation, so that the carbon cell wall of wood uses the carbon–silicon reaction in situ to generate high-strength silicon carbide [28,29]. Simultaneously, nanowhiskers are one-dimensional materials with highly fibrous structures. Through a comparison of different nanowhiskers in the literature, such as oxide nanowhiskers [30], metal nanowhiskers [31], organic nanowhiskers [32], and carbon–silicon nanowhiskers [33], it was found that β-SiC whiskers possess the highest hardness, maximum modulus, maximal tensile strength, and the strongest heat resistance advantages. Moreover, they can be easily compounded with other materials. The incorporation of β-SiC whiskers with a high aspect ratio into the wood–ceramic matrix aims to improve its friction and wear performance due to the whisker pull-out, bridging, and crack deflection mechanisms. The crystallization modification of SiC is primarily achieved by controlling the synthesis conditions and process parameters during the crystal growth process in order to adjust the crystal structure, properties, and applications of silicon carbide. In this study, fir wood was selected as the carbon source. After carbonization, the obtained fir wood carbon can retain the unique wood structure features during the preparation of SiC ceramics, which is advantageous for the infiltration of liquid Si. Using fir as a raw material for preparing C–SiC composites has both considerable economic benefits and complex shape capability. Moreover, the response surface optimization analysis method was employed to determine the optimal preparation process for biomass-based silicon carbide ceramics.

## 2. Materials and Methods

### 2.1. Preparation of Blanks

In this study, the materials of biomass-based silicon carbide ceramic blanks were biomass carbon (fir wood, from the Central South Forestry Technology University, Changsha, China) and silicon powder (27 μm, from Changsha Tianjiu Metal Materials Co., Ltd., Changsha, China), with polyethylene glycol as the adhesive, which was dissolved by heating with anhydrous ethanol (from Shanghai Titan Technology Co., Ltd., Shanghai, China).

Fir wood was used as the raw material, which was first washed with water to remove dirt and impurities and then dried in a drying oven at 80 °C for 24 h to reduce the moisture content. The dried fir wood was ground in a planetary ball mill and sieved with a 500-mesh screen to obtain a powder of 10 g. The sieved fir wood powder was then placed into an argon atmosphere tube furnace and charred at 800 °C, kept warm for 2 h, and then cooled with the furnace. The fir wood powder was ball milled in a planetary ball mill at 200 r/min for 1 h. Finally, the ball-milled fir wood toner was subjected to a second carbonization at temperatures of 1200 °C, 1400 °C, and 1600 °C, then taken out and bagged for use. A mass ratio of 3:7 was used to weigh out the fir-derived carbon powder and silicon powder, which were then ground into a homogeneous mixture with a quartz mortar. Using anhydrous ethanol as the solvent, 5% polyethylene glycol (PEG) by mass of the carbon and silicon mixture was added as a binder and heated to dissolve. The mixture was further ground and dried in an oven at 70 °C for approximately 15 min to make the anhydrous ethanol in the mixture evaporate cleanly. Subsequently, the mixed powder was weighed 2.0 g on an analytical balance and pressed into a 20 mm diameter, 4 mm thick sample using a powder compactor. The parameters of the press were set to 10 MPa, and the pressure was held for 1 min to obtain the FSCC blanks.

### 2.2. Sintering Process of Silicon Carbide Ceramics

The sintering process parameters of the biomass-based silicon carbide ceramics were determined based on references and preliminary experiments [34,35]. The silicon carbide ceramic blanks were placed into a high-temperature carbonization tubular furnace with an argon atmosphere, and to avoid cracking and deformation during carbonization, the temperature was slowly increased during the initial heating. The furnace cavity was raised to the specified temperature using programmed temperature control, first at 5 °C/min to 300 °C, with a controlled heating rate of 10 °C/min between 300 and 1400 °C, and between 1400 °C and 1600 °C with a controlled heating rate of 2 °C/min. The material was held at 1600 °C for 2 h and finally cooled with the furnace, taking out the specimen to obtain the FSCC material.

### 2.3. Characterization and Equipment

The morphology and wear surface of the biomass-based SiC ceramics were examined using a scanning electron microscope (SEM; ZEISS Sigma, Panaco, Groningen, The Netherlands). The phase composition was characterized with X-ray diffraction (XRD; SD/MAX-RB, Rigaku, Tokyo, Japan) using a monochromatic Cu-Kα radiation source. The fir powder was carbonized in a high-temperature carbonization tube furnace (OTF-1200X, GSL-1750X, Hefei Kejing Materials Technology Co., Ltd., Hefei, China), and the SiC ceramics were sintered. The friction and wear properties of the samples were evaluated using a high-temperature friction and wear testing machine (HT-1000, Lanzhou Zhongke Kaihua Science and Technology Development Co. Ltd., Lanzhou, China). The micro-Vickers hardness of the specimens was measured using an HX-1000TM/LCD digital display micro-Vickers hardness tester (Shanghai Optical Instrument Sixth Factory, Shanghai, China), and a metallographic microscope (MR2000, Nanjing Sibo Instrument Technology Co., Nanjing, China) was used to analyze the friction and wear surface.

### 2.4. Response Surface Optimization Design

A Box–Behnken central combination design was employed to investigate the influence of three factors, namely silicon carbide mass ratio, reaction temperature, and adhesive dosage, on the response values of the friction coefficient of the fir-based silicon carbide ceramic material (FSCC) [36,37,38]. Seventeen experimental combinations (two parallel replications for each experimental combination) were generated at three levels for each factor. The results of the friction wear test were collected, and the friction coefficient with similar size of wear rate was selected as the response value under the response surface optimization process conditions. The factor levels and codes are summarized in Table 1, and the experimental protocol and results are presented in Table 2.

The dynamic friction coefficients of the FSCC were analyzed using multi-factor variance analysis in Microsoft Excel and response surface analysis in Design Expert 11.0. The results of the Design Expert 11.0 software design analysis indicated that the response surface regression equations for the dynamic friction coefficients, with respect to the three corresponding factors of carbon to silicon ratio (A), reaction temperature (B), and adhesive dosage (C), are expressed in Equation (1).
Y = 12.78793 − 0.083437A − 0.016632B + 16.53750C + 0.000084AB + 1.23750AC − 0.003625BC − 0.048769A^2^ + 0.00000545812B^2^ − 136.75000C^2^
(1)

An analysis of variance for this equation was conducted, as shown in Table 3.

### 2.5. Analysis of Variance

According to Table 3, the F-value of the quadratic polynomial model is 61.55, and the *p*-value is <0.05, indicating a statistically significant level. The misfit term is greater than 0.05, suggesting that the quadratic polynomial model is a good fit to the data and can accurately reflect the relationship between the dynamic friction coefficient of the FSCC material and the factors. The results indicate that the fitted mathematical model is statistically significant.

The correlation coefficient was R^2^ = 0.9875, indicating a good fit, with 98.75% of the data available for interpretation with this regression model. The predicted R^2^ and the adjusted R^2^ were 0.9023 and 0.9715, respectively, with a difference of less than 0.2. The results showed that the fit between the experimental and predicted values of the dynamic friction coefficient of the fir-based silicon carbide ceramic material was good, and the regression equation could be used to replace the true experimental points and thus fit the effects of various factors. Furthermore, the reaction temperature (B) had the most significant difference, with a *p*-value of less than 0.05, indicating a significant effect on the friction coefficient of the material. However, none of the interaction terms had a significant effect (*p* > 0.05), and the reaction temperature had a significant effect on the friction coefficient in the secondary term (*p* < 0.001).

## 3. Results and Discussion

Based on the data results from Table 3, a quadratic regression fitting was performed, and the response surface interaction plot of the quadratic regression equation was obtained, as shown in Figure 1. The analysis results revealed that the dynamic friction coefficient of FSCC is significantly influenced by various experimental factors. The interactions between the carbon–silicon ratio (A) and reaction temperature (B), as well as reaction temperature (B) and adhesive dosage (C), were more prominent, which is in agreement with the outcomes derived from the analysis of variance.

Figure 1 presents the plane contour plots and three-dimensional response surface plots of the two–two interactions between the three factors: carbon to silicon ratio (A), reaction temperature (B), and adhesive dosage (C). The response surface plot derived from the analysis of variance using the response surface methodology suggests that a flat surface indicates minor effects of factor-level changes on the response value. Conversely, a non-flat surface implies substantial impacts of factor-level changes on the response value, with the response value altering in accordance with the variations in factor levels. The interaction between the two factors can be quantified using the contour plot. When the contour tends to be circular, it indicates that the interaction between the two factors is not significant; conversely, the interaction is significant. The interactive effect of the carbon–silicon ratio and reaction temperature on the friction coefficient of FSCC was investigated through a parabolic distribution, as shown in Figure 1a,d. The friction coefficient of FSCC decreases initially with the increasing carbon–silicon ratio and reaction temperature and then tends to be flat, with the contours tending to be elliptical, indicating a substantial interactive effect between the two factors. Furthermore, the curve of the reaction temperature is significantly steeper than that of the carbon–silicon ratio, indicating that the change in reaction temperature had a more significant influence on the friction coefficient of FSCC.

Figure 1b,e show that the FSCC friction coefficient increases and then decreases with the increase in the carbon to silicon ratio, but the trend is slow. Meanwhile, the FSCC friction coefficient increases and then decreases with the increase in adhesive dosage, and the effect of adhesive dosage on the friction coefficient of FSCC is greater than the effect of the carbon to silicon ratio on its friction coefficient. Figure 1c,f show a parabolic surface distribution of the interaction between reaction temperature and adhesive dosage on the friction coefficient of FSCC, and the friction coefficient of FSCC decreases with the increase in reaction temperature and then tends to be flat. Comparing the absolute values of the primary coefficients of the regression equation, the primary and secondary relationships of the factors influencing the friction coefficients of the materials are determined as B > A > C, i.e., reaction temperature > carbon to silicon ratio > adhesive dosage.

Based on the response surface fitting equation and model prediction results, along with the practicality of actual process settings, the optimal preparation process for achieving the optimum FSCC friction coefficient was determined to involve a carbon to silicon ratio of 3:7, a reaction temperature of 1599.58 °C, and an adhesive dosage of 5.1% of the mixture mass. This resulted in a predicted friction coefficient of 0.361. The reaction temperature was adjusted to 1600 °C, the adhesive dosage was set to 5% of the mass of the carbon and silicon mixture, and three parallel tests were conducted, yielding an average FSCC friction coefficient of 0.373, which closely aligned with the model’s predicted result. The findings of this study indicate that the optimization approach based on response surface model analysis is effective and feasible for the preparation of fir-based silicon carbide ceramic materials, and the model accurately represents the relationship between the FSCC friction coefficient and the selected factors.

The fir-based silicon carbide ceramic material (FSCC) was prepared based on the results of response surface optimization experiments. Figure 2a,b show the XRD patterns of the FSCC intermediate and surface layers, respectively. According to the spectra, the main phases of FSCC are SiC and Si, where all the generated SiC is β-SiC, with corresponding (111) crystal plane line angles of 35.6°, 41.3°, 59.9°, 71.7°, and 75.3°. Figure 2a shows the diffraction peaks of both Si and SiC, where the diffraction backbone is found to be stronger due to the relatively low temperature of the intermediate layer, insufficient sintering, slow reaction of some solid Si + C, and slow diffusion of atoms. Figure 2b shows only the diffraction peak of β-SiC, and the diffraction peak of residual S is not found, with the strongest diffraction peak located at the (111) crystal plane. This indicates that the in situ reaction sintering is sufficient and the crystallinity of SiC is high, thus indicating a high percentage of SiC crystallization in the ceramics.

An analysis of the X-ray diffraction pattern in Figure 2 reveals that biomass-based carbon sources participate in the pyrolysis process, resulting in the formation of SiC nanowhiskers in situ in the interstice of SiC ceramic composite through treatment at 1600 °C. SiC nanowhiskers were synthesized in situ in the interstices of the SiC ceramic composites with treatment at 1600 °C. Figure 3a shows that the surface of the material was uniformly covered with a dense cluster of nanostructures, characterized by SiC nanowhiskers of several microns in length, distributed crosswise and longitudinally without a preferential alignment direction. The Si powder melted rapidly and reacted with a nearby carbon source, generating SiC nuclei in the active position of the powder, which then grew into nanowires along the same direction. This in situ synthesis of β-SiC whiskers was conducted with the purpose of strengthening and toughening the SiC material. In order to gain further insight into the morphology and distribution of the formed nanostructures, a high-magnification scanning electron microscope (SEM) image of the silicon carbide nanowhiskers is presented in Figure 3b. It is observed that the nanostructures mainly consist of nanowhiskers and nanoparticles that coexist in clusters and exhibit high surface roughness. The nanowhiskers are in the form of dendrites and grow in a disordered direction, with conical-like or blunt-headed end shapes, and they serve to connect the unevenly distributed silicon carbide particles, which display high strength and toughness. Furthermore, these nanowhiskers and nanoparticles are interpenetrated to form a three-dimensional network structure, which is more stable than the structure formed by individual nanowhiskers from a structural mechanics perspective [39]. Therefore, the nanostructures presented in this work could be used as toughening reinforcements for different coating materials to compensate for the detrimental effects of the porous structure of biomass. In Figure 3c, scanning electron microscopy (SEM) imaging reveals a more uniform size of SiC particles with a mostly cylindrical shape and a more continuous and denser SiC layer without obvious cavities. Figure 3d shows that the surface of the SiC particles was not smooth but rather composed of many small particles, with a certain number of microcracks present on the SiC layer.

The morphology of the FSCC section combined with the EDS surface scan results in Figure 4 reveals that these nanorods and the whisker structure are mainly composed of Si (69.38%) and C (27.96%) elements, with low oxygen content and uniform distribution of each element. The EDS analysis map further provides the atomic percentages of each element and their atomic percentages at specific positions. Thus, the EDS analysis map constitutes a useful reference for the chemical composition of SiC ceramic materials.

Figure 5 shows the friction coefficient curves of FSCC under different load pressures with wear time. It can be seen that the curves of the friction coefficients can be divided into two periods: the initial break-in period and the smooth wear period. In the initial break-in phase, the friction coefficient rises rapidly due to the engagement, deformation, shearing, and fracturing of microprotrusions on the specimens’ surfaces, while a large number of abrasive particles are generated by the fracture of microprotrusions, resulting in the low-speed performance of the abrasive wear mechanism. The average friction coefficients of FSCC obtained at room temperature under load pressures of 5 N, 10 N, and 15 N were 0.376, 0.477, and 0.582, respectively. This indicates that with increasing load pressure, the overall friction coefficient becomes larger, and the fluctuation of the friction coefficient gradually becomes larger. In the smooth wear period, the point contact friction with Si_3_N_4_ on the grinding ball becomes the surface friction, and the friction coefficient tends to level off. With the increase in load, the friction surface of the high roughness peaks and micro-convex body increase, leading to increased surface roughness and further increased friction stress. At the same time, oxidation of the rough peaks and micro-convexity under the action of load passivation, in combination with the higher load, causes the formation of abrasive grains [27]. These abrasive grains underwent a fine grinding process, resulting in rolling friction under sliding friction, as well as an oxide film layer formed through further oxidation during friction. In addition, adhesion wear gradually replaced abrasive wear, resulting in an increase in the actual contact area. As a result, while the coefficient of friction of the SiC ceramics remained relatively stable, the coefficient of friction of the friction surface increased.

Figure 5b shows that increasing the wear pressure increased the high roughness peaks and micro-convexity of the friction surface, leading to an increase in surface roughness and a decrease in wear rate, as well as an increase in friction stress. The wear rates of FSCC obtained under the load pressures of 5 N, 10 N, and 15 N were 4.64 × 10^−4^ mm^3^/N·m, 6.83 × 10^−4^ mm^3^/N·m, and 7.96 × 10^−4^ mm^3^/N·m, respectively, measured at room temperature. The widths and depths of the FSCC wear marks obtained were 0.767 mm, 1.266 mm, and 1.283 mm and 12.667 μm, 24.88 μm, and 38.254 μm, respectively, as shown in Figure 5c. The raised profile observed at the edges of the substrate wear marks is attributed to the low overall hardness of the material, which leads to plastic flow on the material surface when the frictional heat causes the temperature to rise. This plastic flow facilitates the embedding of grinding balls into the material surface and results in material bulging at the edge of the wear scar. Additionally, the material wear profile under low-pressure wear is significantly smoother.

## 4. Conclusions

In this work, Chinese fir powders were used as the carbon source, which should be subjected to secondary carbonization to the sintering temperature. An in situ reaction between carbon and silicon was conducted to obtain the silicon carbide ceramic. To look for the optimal process as a friction material, a response surface optimization approach proved the preparation of FSCC as following the carbon to silicon ratio of 3:7, a reaction temperature of 1600 °C, and a 5% adhesive dosage. X-ray diffraction analysis could illustrate that the diffraction peaks of SiC were strongest in the (111) crystal plane; at the same time, it could be inferred that the preferential growth trend of the crystal plane was observed. Scanning electron microscopy analysis showed the presence of nanowhiskers attached to the sample’s surface, pores, and granular, which can present SiC particles of relatively uniform size and the compact SiC layer. The friction coefficients of 0.376, 0.477, and 0.582 and the wear rates of 4.64 × 10^−4^ mm^3^/N·m, 6.83 × 10^−4^ mm^3^/N·m, and 7.96 × 10^−4^ mm^3^/N·m, respectively, were inspected for the SiC samples under load pressures of 5 N, 10 N, and 15 N.

## Figures and Tables

**Figure 1 materials-16-04487-f001:**
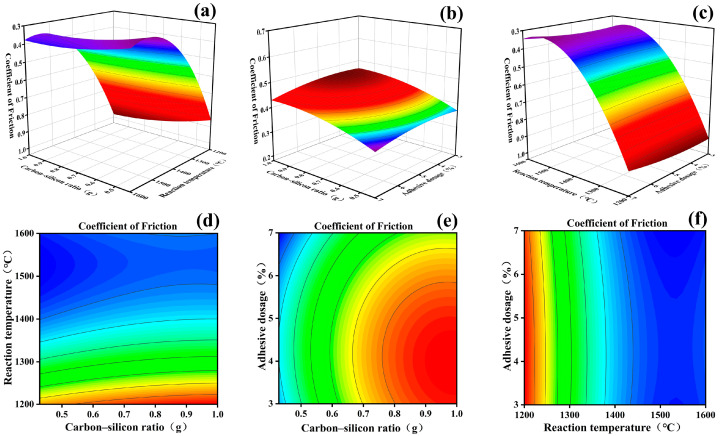
Contour plot of the relation among the carbon–silicon ratio, reaction temperature and adhesive dosage, respectively, 3D diagram (**a**) and plan view (**d**) of the relation between carbon–silicon ratio and reaction temperature, 3D diagram (**b**) and plan view (**e**) of the relation between carbon–silicon ratio and adhesive dosage, 3D diagram (**c**) and plan view (**f**) of the relation between reaction temperature and adhesive dosage.

**Figure 2 materials-16-04487-f002:**
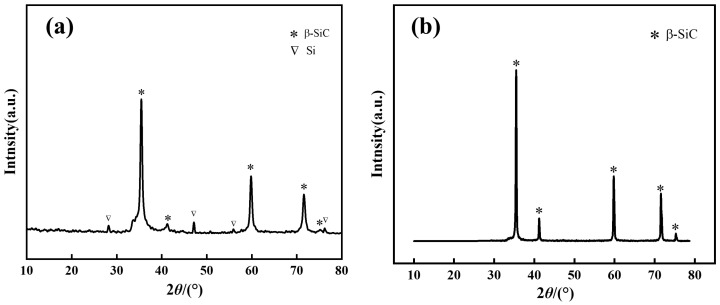
XRD pattern of FSCC powder at different locations: (**a**) Intermediate layer; (**b**) Surface layer.

**Figure 3 materials-16-04487-f003:**
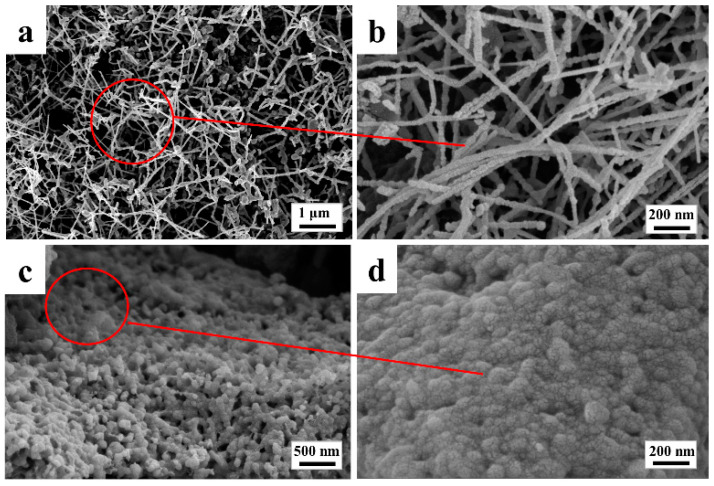
SEM images of FSCC: (**a**,**b**) Surface; (**c**,**d**) Cross-section.

**Figure 4 materials-16-04487-f004:**
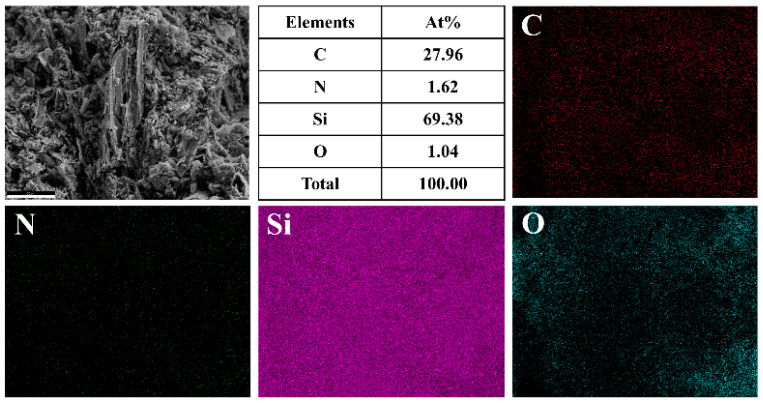
Distribution of EDS elements in FSCC cross-section.

**Figure 5 materials-16-04487-f005:**
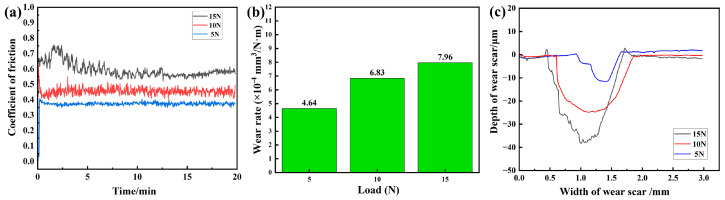
The tribological performance of FSCC under different load pressure tests: (**a**) Friction Coefficient Curves; (**b**) Wear rate; (**c**) Wear profile.

**Table 1 materials-16-04487-t001:** Factor level and coding of response surface experiments.

Horizontal Coding	Carbon–Silicon Ratio (A/g)	Reaction Temperature (B/℃)	Adhesive Dosage (C/%)
−1	3:7	1200	3%
0	4:6	1400	5%
1	5:5	1600	7%

**Table 2 materials-16-04487-t002:** Response surface experiment design and results.

NO.	Carbon–Silicon Ratio (A/g)	Reaction Temperature (B/℃)	Adhesive Dosage (C/%)	Coefficient of Friction (R)
1	−1	−1	0	1.060
2	1	−1	0	0.844
3	−1	1	0	0.442
4	1	1	0	0.342
5	−1	0	−1	0.464
6	1	0	−1	0.394
7	−1	0	1	0.434
8	1	0	1	0.397
9	0	−1	−1	0.945
10	0	1	−1	0.372
11	0	−1	1	0.967
12	0	1	1	0.365
13	0	0	0	0.424
14	0	0	0	0.456
15	0	0	0	0.450
16	0	0	0	0.523
17	0	0	0	0.435

**Table 3 materials-16-04487-t003:** The results of variance analysis of friction coefficient fitting regression equation.

Source	Sum of Squares	df	Mean Square	F-Value	*p*-Value	
**Model**	0.8359	9	0.0929	61.55	<0.0001	**significant**
A-Carbon–silicon ratio	0.0155	1	0.0155	10.26	0.015	
B-Reaction temperature	0.6183	1	0.6183	409.73	<0.0001	
C-Adhesive dosage	0.0001	1	0.0001	0.0119	0.9161	
AB	0.0005	1	0.0005	0.3355	0.5806	
AC	0.0003	1	0.0003	0.1804	0.6838	
BC	0.0002	1	0.0002	0.1393	0.72	
A^2^	0.002	1	0.002	1.31	0.2899	
B^2^	0.2007	1	0.2007	133	<0.0001	
C^2^	0.0008	1	0.0008	0.5218	0.4935	
**Residual**	0.0106	7	0.0015			
**Lack of Fit**	0.0046	3	0.0015	1.02	0.4712	**not significant**
**Pure Error**	0.006	4	0.0015			
**Cor Total**	0.8465	16				

## Data Availability

No new data were created or analyzed in this study. Data sharing is not applicable to this article.

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
