# Peer review of "Fabrication and Process Optimization of Chinese Fir-Derived SiC Ceramic with High-Performance Friction Properties"

_materials, 2023, doi:10.3390/ma16124487_

Round 1

Reviewer 1 Report

I have reviewed the manuscript entitled "Fabrication and Process Optimization of Chinese Fir Derived SiC Ceramic with High-performance Friction Property" written by Liu et. al. Although the manuscript shows significant interest to the readers, I would like to suggest/recommend a few points as follows:

1) Introduction

Page 2, Line 76:

Please add 2-3 paragraphs of literature review on the fir-derived carbon and response surface analyses which are the main subject of this study.

2) Materials and Methods

Page 2, Lines 92-93:

The authors should specify or state the proportion and the mass fraction used.

3) Results and Discussion

Page 7; Lines 237-238

Which characterization does the author refer to observe the destruction of the tubular structure if the SEM image fails to do that analysis? 

Page 7; Lines 262-263

Which data show the difference in expansion coefficient?

Page 9; Lines 288-292

Which theory or model do the authors refer to claim these points?

Author Response

Point 1: Introduction

Page 2, Line 76: Please add 2-3 paragraphs of literature review on the fir-derived carbon and response surface analyses which are the main subject of this study.

Response 1:

Thanks for your comments on our paper. We have added to the abstract section based on your suggestions, as shown below.

In this study, fir wood was selected as the carbon source. After carbonization, the obtained fir wood carbon can retain the unique wood structure features during the prep-aration of SiC ceramics, which is advantageous for the infiltration of liquid Si. Using fir as raw materials for preparing C–SiC composites has both considerable economic ben-efits and complex shape capability. Moreover, the response surface optimization anal-ysis method was employed to determine the optimal preparation process for bio-mass-based silicon carbide ceramics.

Point 2: Materials and Methods

Page 2, Lines 92-93: The authors should specify or state the proportion and the mass fraction used.

Response 2:

Thank you for your comments on our paper. We have added to the Materials and Methods section based on your suggestions, as shown below.

A mass ratio of 3:7 was used to weigh out fir-derived carbon powder and silicon pow-der, which were then grounded into a homogeneous mixture with a quartz mortar. Using anhydrous ethanol as the solvent, 5% polyethylene glycol (PEG) by mass of the carbon and silicon mixture was added as a binder and heated to dissolve.

Point 3: Results and Discussion

  • Page 7; Lines 237-238: Which characterization does the author refer to observe the destruction of the tubular structure if the SEM image fails to do that analysis?

Response (1):

Thank you for your comments on our paper. We have revised and added to the Results and Discussion section based on your suggestions, as shown below.

The analysis of the X-ray diffraction pattern in Figure 2 has revealed that bio-mass-based carbon sources participate in the pyrolysis process, resulting in the for-mation of SiC nanowhiskers in situ in the interstice of SiC ceramic composite through treatment at 1600°C.

  • Page 7; Lines 262-263: Which data show the difference in expansion coefficient?.

Response (2):

Thank you for your comments on our paper. In response to your suggestions, we would like to respond to the results and discussion section:

In response to your suggestions, we have discussed and deliberated on the decision to delete this section due to the lack of verbal accuracy here. In doing so, we have carefully considered the impact of these changes on the overall quality of the article and have tried to maintain the coherence and integrity of the content of the manuscript. We hope that these changes will meet your expectations and bring the manuscript more in line with the requirements of the journal.

Thank you again for your time and patience, and we look forward to your further review of the revised manuscript and hope to continue to receive your valuable comments.

  • Page 9; Lines 288-292: Which theory or model do the authors refer to claim these points?

Response (3):

Thank you for your comments on our paper. In response to your suggestion, we have added literature support to this proposed friction process to assert this argument:

  1. Sharma, S.K.; Manoj Kumar, B.V.; Kim, Y.-W. Tribology of WC reinforced SiC ceramics: Influence of counterbody. Friction 2018, doi:10.1007/s40544-017-0194-2.

Reviewer 2 Report

 It would be helpful to provide a section outlining the motivation, objectives, and significance of the study in the introduction part.

Author Response

Thanks for your great jobs.

Reviewer 3 Report

Although this topic is of some interest, this manuscript in its present form cannot be recommended for publication and requires some improvement and clarification.

1.     From the Introduction it is not clear what crystalline modification of silicon carbide is meant.

2.     The abstract reports on nanowhiskers, however  in the introduction  no words is said about these and other nanostructures.

3.     Line 55, where “ SiC ceramics are used …”.  Here it will be important to mention other important applications in optical devices, nanotechnology and nuclear and space material science. This is important to attract more reader interest and further incentive applications. For some of them, see, for example:

a)     Tynyshbayeva, K.M.;  et al. Study of Helium Swelling and Embrittlement Mechanisms in SiC Ceramics. Crystals 202212, 239. https://doi.org/10.3390/cryst12020239

b)    Huczko, A., Dąbrowska, A., et al. Silicon carbide nanowires: synthesis and cathodoluminescence. physica status solidi (b), 2009, 246(11‐12), 2806-2808. https://doi.org/10.1002/pssb.200982321

c)     Lebedev, A. S.,  et al (2020). Carbothermal Synthesis, Properties, and Structure of Ultrafine SiC Fibers. Inorganic Materials56, 20-27.

4.     Line 224. I wonder why other phases are not formed? Is it possible that they are formed, but their concentration is so low that it is not visible in diffraction, but can be seen in Roman or luminescence?

5.     It is not entirely clear how the porosity was analyzed?

6.     In Conclusion, it would be very useful to see a clearer formulation of what new data about SiC were obtained in this work.

In general, the manuscript is interesting and can be considered for publication after constructive reflection on the above comments.

Author Response

Point 1: From the Introduction it is not clear what crystalline modification of silicon carbide is meant.

Response 1:

Thank you for your comments on our paper. In response to your suggestions, we have made changes and additions to the definition of whisker modification in the abstract section, as follows:

The incorporation of β-SiC whiskers with a high aspect ratio into the wood-ceramic matrix aims to improve its friction and wear performance due to the whisker pull-out, bridging, and crack deflection mechanisms. The crystallization modification of SiC is primarily achieved by controlling the synthesis conditions and process parameters during the crystal growth process in order to adjust the crystal structure, properties, and applications of silicon carbide.

Point 2: The abstract reports on nanowhiskers, however  in the introduction  no words is said about these and other nanostructures.

Response2:

Thank you for your comments on our paper. In response to your suggestions, we have made changes and additions to the abstract section regarding the introduction of nanowhiskers, as follows:

Simultaneously, nanowhiskers are one-dimensional materials with highly fibrous structures. Through the comparison of different nanowhiskers in the literature, such as oxide nanowhiskers , metal nanowhiskers , organic nanowhisk-ers , and carbon-silicon nanowhiskers , it is found that β-SiC whiskers possess the highest hardness, maximum modulus, maximal tensile strength, and the strongest heat resistance advantages. Moreover, they can be easily compounded with other materials. The incorporation of β-SiC whiskers with a high aspect ratio into the wood-ceramic matrix aims to improve its friction and wear performance due to the whisker pull-out, bridging, and crack deflection mechanisms.

Point 3: Line 55, where “ SiC ceramics are used …”.  Here it will be important to mention other important applications in optical devices, nanotechnology and nuclear and space material science. This is important to attract more reader interest and further incentive applications.

Response3:

Thank you for your comments on our paper. In response to your suggestions, we have made changes and additions to the literature in this section, as follows:

  1. Lebedev, A.S.; Suzdal’tsev, A.V.; Anfilogov, V.N.; Farlenkov, A.S.; Porotnikova, N.M.; Vovkotrub, E.G.; Akashev, L.A. Carbothermal Synthesis, Properties, and Structure of Ultrafine SiC Fibers. Inorganic Materials 2020, 56, 20-27.
  2. Tynyshbayeva, K.M.; Kadyrzhanov, K. K, et al. Study of Helium Swelling and Embrittlement Mechanisms in SiC Ce-ramics. Crystals 2022, 12, 239. https://doi.org/10.3390/cryst12020239.
  3. Huczko, A.; Dąbrowska, A.; Savchyn, V.; Popov, A.I.; Karbovnyk, I. Silicon carbide nanowires: synthesis and cathodoluminescence. physica status solidi (b) 2009, 246, 2806-2808.

Point 4: Line 224. I wonder why other phases are not formed? Is it possible that they are formed, but their concentration is so low that it is not visible in diffraction, but can be seen in Roman or luminescence?

Response4:

We are very grateful to you for reviewing our paper and providing valuable comments during your busy schedule. In response to your question about only β-SiC phase being observed in the XRD results and no other phases are observed, we would like to reply as follows:

In our study, the experimental process did focus on whether other phases would form, but in the XRD patterns obtained, we only observed characteristic peaks for the β-SiC phase and not for the other phases. When preparing the SiC ceramic material, we chose high temperature sintering conditions of 1600°C to ensure that the β-SiC phase was mainly formed during the experiments and to suppress the formation of other phases as much as possible. Secondly, β-SiC has a higher stability compared to other possible phases, and thus the characteristic peaks of the β-SiC phase were mainly observed in the XRD patterns. Finally, we conducted several experiments and never observed the characteristic peaks of other phases during the experiments. This also increases our confidence that the sample is mainly composed of β-SiC phase.

Point 5: It is not entirely clear how the porosity was analyzed?

Response5:

We are very grateful for your careful review of our paper and your valuable comments. In response to your question about the analytical porosity, we would like to respond as follows:

In this study, we focused on the preparation process, microstructure and frictional wear properties of SiC ceramic materials and presented the experimental results systematically in the paper. Indeed, the analysis of porosity is not addressed in our study at this stage. We acknowledge that porosity is an important parameter in some respects and that it may have a significant impact on the mechanical properties, electrical conductivity, etc. of the material. However, in the context and purpose of this paper, our main concern is the preparation of SiC ceramic materials and their correlation with frictional wear performance parameters, which we have achieved in the related discussions and experimental data analysis. Considering factors such as experimental resources and lead time, we are unable to add experimental analysis about porosity for the time being. However, we appreciate your suggestion and in our future research, we will focus on the influence of porosity and the corresponding analysis methods to investigate the correlation between porosity and material properties in depth.

Point 6: In Conclusion, it would be very useful to see a clearer formulation of what new data about SiC were obtained in this work.

In general, the manuscript is interesting and can be considered for publication after constructive reflection on the above comments.

Response6:

Thank you verymuch. We have tried our best to revise our paper according to the reviewers’s advices. We have carefully deliberated the results and discussions again.

Round 2

Reviewer 3 Report

The authors have successfully improved their original manuscript, so now it can be recommended for publication.